# Survival and Proliferation under Severely Hypoxic Microenvironments Using Cell-Laden Oxygenating Hydrogels

**DOI:** 10.3390/jfb12020030

**Published:** 2021-05-02

**Authors:** Shabir Hassan, Berivan Cecen, Ramon Peña-Garcia, Fernanda Roberta Marciano, Amir K. Miri, Ali Fattahi, Christina Karavasili, Shikha Sebastian, Hamza Zaidi, Anderson Oliveira Lobo

**Affiliations:** 1Division of Engineering in Medicine, Department of Medicine, Brigham and Women’s Hospital, Harvard Medical School, Cambridge, MA 02139, USA; erikberivan@gmail.com (B.C.); akmirir@gmail.com (A.K.M.); a.fatahi.a@gmail.com (A.F.); karavasc@pharm.auth.gr (C.K.); sebastian.shikha@gmail.com (S.S.); zaidi.hamza@gmail.com (H.Z.); 2Materials Science and Engineering Graduate Program, UFPI-Federal University of Piaui, Teresina 64049-55, PI, Brazil; ramon.raudel@ufrpe.br (R.P.-G.); marciano@ufpi.edu.br (F.R.M.); 3Academic Unit of Cabo de Santo Agostinho, Federal Rural University of Pernambuco, Cabo de Santo Agostinho 52171-900, PE, Brazil; 4Department of Physics, Federal University of Piaui, Teresina 64049-550, PI, Brazil; 5Biofabrication Lab, Department of Mechanical Engineering, Rowan University, Engineering Hall, Glassboro, NJ 08028, USA; 6Center for Applied NanoBioscience and Medicine, College of Medicine-Phoenix, University of Arizona, Phoenix, AZ 85004, USA; 7Department of Pharmaceutical Technology, School of Pharmacy, Aristotle University of Thessaloniki, GR-54124 Thessaloniki, Greece; 8LIMAV-Interdisciplinary Laboratory for Advanced Materials, UFPI-Federal University of Piaui, Teresina 64049-550, PI, Brazil

**Keywords:** oxygen, sever hypoxia, microparticles, electrospray, hydrogels, cells, biomaterials

## Abstract

Different strategies have been employed to provide adequate nutrients for engineered living tissues. These have mainly revolved around providing oxygen to alleviate the effects of chronic hypoxia or anoxia that result in necrosis or weak neovascularization, leading to failure of artificial tissue implants and hence poor clinical outcome. While different biomaterials have been used as oxygen generators for in vitro as well as in vivo applications, certain problems have hampered their wide application. Among these are the generation and the rate at which oxygen is produced together with the production of the reaction intermediates in the form of reactive oxygen species (ROS). Both these factors can be detrimental for cell survival and can severely affect the outcome of such studies. Here we present calcium peroxide (CPO) encapsulated in polycaprolactone as oxygen releasing microparticles (OMPs). While CPO releases oxygen upon hydrolysis, PCL encapsulation ensures that hydrolysis takes place slowly, thereby sustaining prolonged release of oxygen without the stress the bulk release can endow on the encapsulated cells. We used gelatin methacryloyl (GelMA) hydrogels containing these OMPs to stimulate survival and proliferation of encapsulated skeletal myoblasts and optimized the OMP concentration for sustained oxygen delivery over more than a week. The oxygen releasing and delivery platform described in this study opens up opportunities for cell-based therapeutic approaches to treat diseases resulting from ischemic conditions and enhance survival of implants under severe hypoxic conditions for successful clinical translation.

## 1. Introduction

Higher-level organisms have evolved advanced circulatory systems that provide tissues with nutrients, e.g., glucose and oxygen via blood. Any aberration in this system can impair the delivery of nutrients and result in starvation-induced cell death, causing permanent loss of tissue function [1]. This has resulted in an increasing demand to generate clinical sized vascularized tissue-engineered constructs that can be implanted to alleviate anoxia induced tissue loss [2]. However, fabricating thick tissues in the lab poses challenges from the perspective of mass transport and diffusion limitations. A new class of biomaterials that release oxygen have been proposed and is being investigated to resolve these issues in lab fabricated tissue-like constructs [1,3]. Oxygen releasing materials aim to sustain or restore a tissue’s aerobic respiration, which drives carbohydrate metabolism and all the downstream physiologic functions. These biomaterials have been shown to improve cell survival under hypoxic conditions where oxygen supply is limited [4]. Maintaining mild hypoxia under anoxic conditions can promote vascularization and angiogenesis. This is critical for various regenerative engineering applications, especially tissue formation and healing processes [5]. Two critical factors that determine the angiogenic effects from any oxygen releasing material are duration for which oxygen is released and the concentration that is maintained during its release. Oxygen more than normoxic condition is equally detrimental as when it is below a certain amount that leads to severe hypoxia [6,7]. As such, maintaining it within a particular concentration regime over a certain period of time necessitates its angiogenic and vasculogenic properties. While different biomaterials have been proposed and studied in the form of thin-films, hydrogen peroxide laden microparticles, perfluorocarbons, solid peroxides, etc., most of these platforms have offered less freedom in terms of oxygen release payloads or duration of release, leading to unfavorable outcomes [8].

Here, we present an engineered system consisting of oxygen releasing microparticles (OMPs) and explore these to sustain aerobic respiration of artificial tissues over a period of approximately 2 weeks. Chemical species such as calcium peroxide (CPO)-based oxygen generation have been exploited sustaining the cells’ metabolic needs under anoxic conditions; however, the unrestrained oxygen generation and release detrimentally affected the consequent oxygen tensions’ stability [5,9]. In its essence, while oxygen converts nutrients into energy; however, as pointed out before, it is highly important to maintain its concentration and release duration within a specific window. CPO is known for its burst release [10]. CPO may be able to support oxygen release at rates and durations that are angiogenic and enhance survival and neovascularization, only if its release is properly regulated. As the oxygen generation capability of CPO is based on hydrolysis, restricting water molecules’ availability can control the subsequent oxygen production from CPO and hence its release duration. Encapsulation of CPO in a hydrophobic polymer has been shown to extend release times and also lower the toxicity from its ROS that are formed as by-products during its hydrolysis [11]. Polycaprolactone (PCL) encapsulation results in controlling hydrolysis of CPO by limiting water access and hence leads to sustained oxygen release (Figure 1). We studied cell survival of cells encapsulated within OMP containing gelatin methacryloyl (GelMA) hydrogels under severely hypoxic conditions. The oxygen generation sustained metabolic activity and proliferation rates of encapsulated cells. We further tested survival of cells within 3D printed fluidic channels within a Pluronic F-127 (PF-127) made hollow channels as a model of an endothelial vessel.

## 2. Materials and Methods

### 2.1. Materials

Polycaprolactone (PCL, 45 kDa), calcium peroxide (CPO, 200 mesh), PF-127, and polyethylene glycol (PEG400) were obtained from Sigma-Aldrich (St. Louis, MO, USA) and used as received. Chloroform and dichloromethane (DCM) solvents were also purchased from Sigma-Aldrich (USA). Gelatin from porcine skin (type-A, 300 bloom), methacrylic anhydride, 2-hydroxy-4′-(2-hydroxyethoxy)-2-methylpropiophenone (photoinitiator, PI, Irgacure 2959), Triton X-100, and bovine serum albumin (BSA) were also purchased from Sigma-Aldrich (USA). Deionized water (DI water) was taken from a Millipore Milli-Q Reference ultra-pure water purifier (St. Louis, MO, USA). All chemicals were of the analytic grade and used without further treatment. Sylgard 184 Silicone Elastomer kit was purchased from Dow Corning Corporation (Midland, MI, USA), and PMMA sheets were obtained from McMaster-Carr (Elmhurst, IL, USA). Dulbecco’s phosphate-buffered saline (DPBS), fetal bovine serum (FBS), trypsin-ethylenediaminetetraacetic acid (trypsin-EDTA), penicillin/streptomycin, 4′,6-diamidino-2-phenylindole (DAPI), Live/Dead Viability Kit, Alexa Fluor 594- and Alexa Fluor 555-conjugated secondary antibodies were purchased from Abcam (Cambridge, MA, USA). All other chemicals used in this study were obtained from Sigma-Aldrich unless otherwise noted.

### 2.2. Fabrication and Characterization of Oxygen Releasing Microparticles (OMPs)

OMPs were obtained via electrospray (ES) method (Figure 1ai). Briefly, PCL (10% *w/v*) and CPO (1% *w/v*) were dissolved in chloroform by stirring (IKA^®^ RCT essential, San Diego, CA USA) for at least 3 h at room temperature to create a homogenous solution, which was subsequently used for the ES process. PEG400 (0.5% *w/v*) or Pluronic (0.5%) as stabilizer were prepared in DCM and stirred for 1 h. The working temperature was 18 ± 1 °C, and the relative humidity was controlled in the range of 50–60%. To synthesize OMPs with different CPO concentrations and different stabilizing agents, polymer solutions containing 10% PCL, with CPO at 0, 0.5, 1, 2, and 5% (*w/v*):PEG 0.5% and Pluronic 0.5% were used. The respective solutions were poured in a syringe and allowed to flow through a needle (26 gauge) at a feeding rate of 4.2 mL·h^−1^ into a metallic stainless-steel needle (inner diameter: 0.8 mm; outer diameter: 1 mm). ES was enabled by applying a controlled electrical field (Test Equipment, Gainesville, GA, USA, USA) to the processing head (16 kV). Under the action of an optimized electrical force, fine droplets resulting from the ejected fluid jet (under cone-jet mode) were collected at a distance of 15 cm on aluminum foil. After the ES was complete, absolute ethanol was pipetted onto the aluminum foil and sprayed solidified particles were collected directly in a 50 mL falcon tube. The solidified OMPs were put in a desiccator for one week to ensure complete dryness before further characterization. For microscopy, OMPs obtained directly from the desiccator were used for imaging

### 2.3. Size and Morphology of OMPs

Size and surface morphology of OMPs were analyzed using Scanning Electron Microscope (SEM). Briefly, dry OMPs were placed on double-sided black carbon tape. Samples were sputter coated with a 10 nm gold layer under vacuum and examined with SEM (Zeiss Supra55) at 5 kV accelerating voltage.

### 2.4. Release Kinetics of Peroxide (H_2_O_2_) and Oxygen (O_2_) from OMPs

Kinetic assay for H_2_O_2_ release from the OMPs was carried out by H_2_O_2_ Amplex^®^ Red Assay Kit using a spectrophotometer (571/585 nm) and following the manufacturer’s protocol. The reaction follows Michaelis–Menten kinetics. Kinetic parameters *K*_m_ and *V*_max_ were obtained by linear regression analysis of the double reciprocal Lineweaver–Burk plots. 

The turnover number, K_m_ was calculated according to the following equation:1/V = K_m_/V_max_[*S*] + 1/V_max_(1)
where [*S*] is the final concentration of the released H_2_O_2_.

O_2_ release from OMPs was measured using a commercial oxygen sensor (Unisense; Denmark). OMPs at different concentrations were put in anoxic phosphate buffered saline (PBS) and studied for oxygen release every day under a nitrogen gas purged chamber. The sensor was calibrated with a 2-point calibration for PBS with 0 (N_2_ gas purged) and 21% (air saturated) O_2_ concentrations, respectively.

### 2.5. GelMA Synthesis

GelMA was prepared according to an established protocol [12]. Briefly, gelatin from porcine skin was dissolved in PBS for 2 h at 60 °C under constant stirring to yield a 10 % (*w/v*) gelatin solution. To modify with methacryloyl substitution, a volume corresponding to 4% (*v/v*) of methacrylic anhydride was added gradually to the gelatin solution and subsequently incubated and stirred for 1 h at 60 °C and 500 rpm. Two volumes of pre-warmed (37 °C) PBS were then added, and the solution was dialyzed for at least 5 days against deionized water. After dialysis, the GelMA solution was filtered and freeze-dried.

### 2.6. Cell Culture

Cells used for the characterization with the microfluidic system were immortalized skeletal myoblasts, C2C12. The C2C12 cell line was obtained from ATCC^®^ CRL-1772™ (Manassas, VA, USA). Before cell seeding, C2C12 cells were cultured in DMEM cell culture medium supplemented with 10% FBS and 1% penicillin-streptomycin at 37 °C and 5% CO_2_. Media was changed every 2–3 days, and once 80% confluency was reached, they were harvested and used for further steps.

### 2.7. Microfluidic Channel

Microfluidic perfusion channel was fabricated by casting PDMS around laser cut PMMA molds. 4 mm thick PMMA layers (8560K239, McMaster) were machine cut using a laser cutter (Universal Laser Systems, Scottsdale Arizona USA). A bioprinter (BioBots, Inc. Philadelphia, PA, USA) equipped with a 27G blunt needle was used to create a 3D sacrificial scaffold from 40% (*w/v*) PF-127 in water. To create cellularized micro channels, 5% (*w/v*) GelMA in FBS was mixed with 0.2% (*w/v*) 2-hydroxy-4′-(2-hydroxyethoxy)-2-methylpropiophenone. C2C12 cells and OMPs were suspended in this GelMA solution at a final concentration of 3 × 10^6^ cells mL^−1^ and 0–10% (*w/v*) OMPs. Following overnight dehydration, the printout was placed on top of a mold of 10 mm (length) × 4 mm (width) × 4 mm (height) size which was then filled with GelMA prepolymer and cured on both sides under UV (power: 850 mW, OmniCure Series 2000, Heerbrugg, Switzerland). After crosslinking of GelMA, the block was placed in cold PBS and left at 4 °C for 15 min. The liquified sacrificial scaffold of PF-127 was then removed using a syringe. Cell-laden and OMP containing blocks were then placed back inside a tissue culture incubator set at 37 °C for subsequent culture. DMEM was perfused within the straight channels of 500 µm diameter at a continuous flow rate of 20 µL/min for 7 days via a syringe pump (Harvard Apparatus, Holliston, MA, USA).

### 2.8. Cell Viability and Morphology Aanalyses

Cell viability was assessed by a LIVE/DEAD staining assay for live (calcein-AM; Green) and dead (ethidium homodimer-1; Red) cells. Constructs were washed twice with PBS, followed by incubation for 15 min in the live/dead assay solution in the dark. The samples were washed again with PBS and immediately imaged using a Zeiss Imager (Zeiss, Oberkochen, Germany) fluorescent microscope. For proliferation, C2C12 cells were fixed in a 4% formaldehyde solution in PBS. Alexa-568 phalloidin was used to stain f-actin (1:100 dilution). C2C12 was stained using Alexa-488 goat anti-mouse secondary antibody (Invitrogen, 1:200 dilution). Nuclei were counter-stained with DAPI. The staining results were examined using a fluorescent microscope (Zeiss, Germany).

### 2.9. Statistical Analysis

Statistical analysis was performed using Prism 7 (GraphPad Software, Northside Dr., San Diego, CA, USA). A value of *p* < 0.05 was defined as statistically significant. One-way analysis of variance (ANOVA) was used to analyze the (%) cell viability data.

## 3. Results and Discussion

CPO as a solid peroxide has a capability to release oxygen upon hydrolysis. Figure 1a shows the schematics of OMP synthesis, O_2_ release from these particles, and the chemistry that lies behind this chemical process. Producing microparticles using ES is not new and has already been employed to synthesize a wide variety of materials. We used this technique to produce OMPs with CPO as the main oxygen generating source encapsulated in PCL. Compared to CPO concentrations varying from 0.5 to 5% did not seem to influence the size of the OMPs as much as was seen with using the surfactants, PEG or PF-127 at 0.5% (*v/v*). The reason for this observation might be the change in electrostatic and weak intermolecular hydrophilic interactions that groups such as PEG and PF-127 affect or induce via their amphiphilic nature making it possible to increase hydrophilicity of particles encapsulated in an otherwise hydrophobic polymer, such as PCL (representative images in Figure 2). Interestingly, these stabilizing groups also affect the way OMPs interact with cells and thus have been shown to increase cytocompatibility in our particles. Figure 1ai shows the schematics of the synthesis process and SEM images in Figure 2 show the effect these stabilizing agents have on the size of the synthesized OMPs. All the different synthesized particles showed a round shape. According to Montesdeoca et al. [13], indeed porosity and hence access to water for hydrolysis is affected by the PCL encapsulation together with CPO concentration and hence contributes to the oxygen release for a longer time. In addition, it seems that the stabilizing molecules in the form of surfactants that act as capping layer for these OMPs play a pivotal role in defining the degree of diffusion of the medium, facilitating the exchange of nutrients and oxygen for the cells. In the samples with PCL 10% and PF-127 0.5% (Figure 2c.2) there is a slight indication for the production of nanofibers. This is a further indication the effect these stabilizing groups can have on hydrophobic polymer, bringing a change in the overall output in the form of the type of species one can produce—particles or fibers. This result could be associated with selected applied electrical field and solution parameters during the ES process as previously described from our group elsewhere [13,14]. Particle size distribution analysis indicated that all samples have average sizes between 1 and 10 μm. This result is interesting because, as reported elsewhere, regardless of the concentration of the dopant, difference in electric potential and surface viscosity are not affected [15] when prepared under similar synthesis conditions. Similar to what is observed in our study.

We engineer and explore OMPs which can sustain aerobic respiration of cells and possible tissues under severely hypoxic or anoxic conditions. The innovation of oxygen-generating biomaterials is a promising method in the area of biomaterial-based therapies. Approaches such as CPO-based oxygen generators showed the capacity to satisfy the metabolic needs of cells under anoxic conditions; however, their unrestrained oxygen generation or peroxide release had a negative effect on the equilibrium of the resultant oxygen tension or peroxide toxicity, respectively [1,16]. We provide controlled oxygen release that can affect the overall metabolism of the cells. As CPO’s oxygen generation capability is based on hydrolysis, restricting water availability can control this process which in turn can be used to tune oxygen release [17]. Encapsulation of CPO in a hydrophobic polymer extends release times and also lowers the toxicity of CPO for cells. Despite yielding promising results, all tested approaches could only generate oxygen for a short period. Specifically, conventional approaches did not produce a relatively stable oxygen tension. 

Figure 3 shows a clear change in the hydrolysis pattern in presence and absence of PCL. Clearly, PCL encapsulation extended H_2_O_2_ and O_2_ release profiles as compared to when CPO was directly studied in PBS with unhindered hydrolysis. 

Sustained oxygen generation over more than a week’s time positively affects metabolic activity and proliferation rates of cells encapsulated within hydrogels to fabricate artificial tissues or tissues for implants. Figure 4a shows the basic principle and fabrication steps for a microfluidic chip consisting of a hollow vessel and a respective custom-designed PDMS microfluidic device. A PF-127 channel as a sacrificial channel can be dissolved to fabricate a hollow space within a GelMA hydrogel containing construct as shown in Figure 4b. Media could be flown into the device via these hollow channels through the top inlet and leave via the bottom outlet. C2C12 myoblast cells encapsulated within the hydrogel (Figure 5a) showed good viability for the different concentrations of the OMPs co-encapsulated within these hydrogels together with the cells. CPO encapsulation in the hydrophobic polymer, PCL, extended oxygen release times and lowered the toxicity of the CPO (Figure 5b). 

Additionally, our engineering approach for fabricating a hollow vessel by integrating microchannel structures within a crosslinked hydrogel. We printed our channels with PF-127. It is known that PF-127 forms a shear-thinning hydrogel at room temperature and due to its thermoreversible behavior can return to a solution phase at a low temperature and change into a gel when the temperature is increased [18]. A 40% PF-127 aqueous solution was extruded through a syringe needle. The hydrogel retained its shape, allowing for cylindrical channels within a GelMA block. Accordingly, we fabricated GelMA constructs containing a series of microchannels with varied diameters to indicate different blood vessel cross-sections. Our system showed good perfusion performance for over a week using a syringe pump equipped with 3 mL syringes connected to the sample blocks at a flow rate of 1 mL h^−1^. This flow rate was chosen to resemble the physiological conditions. Cell viability was then assessed using live/dead staining, calculated from the ratio between the number of cells treated with the OMPs and non-treated cells (control). The viability of the cells indicated that OMPs showed less to no cytotoxicity.

Although in some cases the cell viability decreased when the concentration of the OMPs increased beyond 3% (Figure 5c), we believe this could be because of increased peroxide concentration or a change in porosity or mechanical properties of the hydrogel that could have been influenced due to either the peroxide or oxygen release. More studies need to be done to understand the chemistry behind such an observation. At 3% OMP, cell viability on day 5 was higher than the other OMP concentrations used. OMPs at higher concentrations decreased cell viability over more than 5 days of the cell culture. This could be stemming from hyperoxia or higher ROS production that affects cell growth, viability, and other cellular functions.

The effect on morphology and cell proliferation was studied using F-actin/DAPI staining of the encapsulated cells within OMPs at concentrations of 1–10% for 7 days under hypoxia (Figure 5c). As expected, no noticeable cell spreading was found on day 1, and all the cells still maintained a round morphology. Cells gradually spread out initially from spindle shapes to later long stripe-like forms over a week. This indicates a better microcellular environment produced as a combination of OMPs’ oxygenation sustaining cellular metabolism over the course of study. The proliferation of the encapsulated cells was also clearly noticeable. Further studies are needed to look at the effect such a microenvironment created by OMPs can have over VEGF production, proliferation or neovascularization that can in turn affect hydrogel degradation and pave a path for eventual anastomosis in an animal study. The degradation profiles also showed the bio-printed vascular structures steadily degraded in the culture medium depending on the incubation period.

## 4. Conclusions

Oxygen-containing biomaterials with outstanding characteristics and benefits in different fields are attractive. We developed an oxygenating platform which acts as a good oxygen source to maintain healthy cellular function, including under ischemic conditions. CPO as a solid peroxide was used as an oxygen source that hydrolyses into oxygen upon contact with water. With unhindered hydrolysis, CPO exhausts its oxygen releasing capability in few days’ time. However, encapsulation within a hydrophobic polymer, such as PCL cuts water access, decreasing hydrolysis and eventually increasing oxygen release times. Microencapsulation of CPO in PCL micromaterials thus allows for safely increasing both CPO payload and oxygen release duration. Such a system can provide oxygen supplies over a prolonged time and can be used in conditions that are affected by ischemia or low oxygen tension. Cell survival was tested in hydrogel constructs of GelMA hydrogel in the presence of OMPs with hollow vessels through which media could be flown in a severely hypoxic environment. The geometry was chosen to recapitulate physiological conditions. It is of note that the current study is limited to the use of PCL. However, it is anticipated that alternative hydrophobic materials can offer similar extended oxygen generation profiles owing to their ability to modify CPO hydrolysis. In nutshell, our system represents a platform to engineer self-oxygenation of tissues, which allows for controlling the in vivo oxygen tension for prolonged periods to enhance the success of implant survival and host acceptance metabolically. It thus represents a good steppingstone towards the development of clinically sized tissues for regenerative applications. 

## Figures and Tables

**Figure 1 jfb-12-00030-f001:**
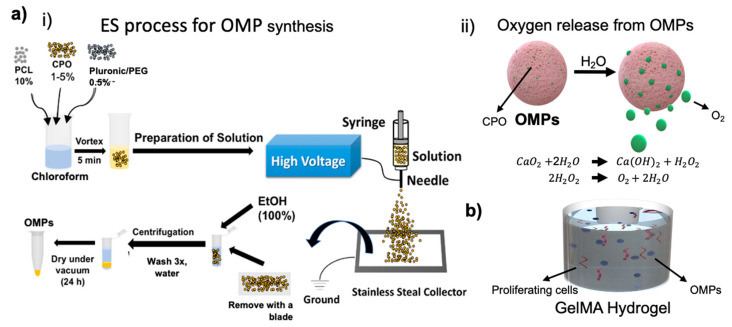
(**ai**) Schematics showing flow chart of the electrospraying (ES) method for synthesis of OMPs, (**aii**) O_2_ release from OMPs upon hydrolysis, (**b**) OMPs in GelMA lead to survival and proliferation of cells.

**Figure 2 jfb-12-00030-f002:**
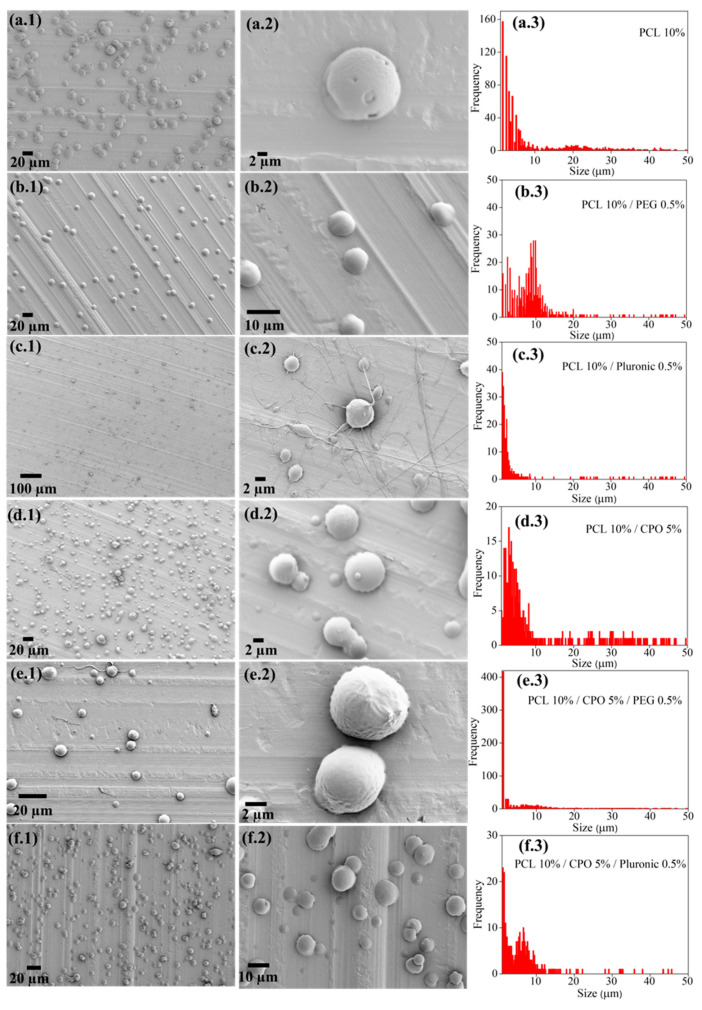
Representative SEM and particle size analysis from OMPs. Controls (without CPO): (**a1–3**) PCL 10%, (**b**) PCL 10%/PEG 0.5% and (**c**) PCL 10%/Pluronic 0.5%: (**d**) PCL 10%/CPO 5%, (**e**) PCL 10%/CPO 5%/PEG 0.5%, and (**f**) PCL 10%/CPO 5%/Pluronic 0.5%. Photos showing particles in low and high magnification showing distribution and surface morphology of the particles (1 and 2); (3) Particle size analysis showing distribution of obtained OMPs. OMPs with only CPO at 5% are presented as the size did not seem to vary with increased CPO concentration but correlated with the stabilizer content.

**Figure 3 jfb-12-00030-f003:**
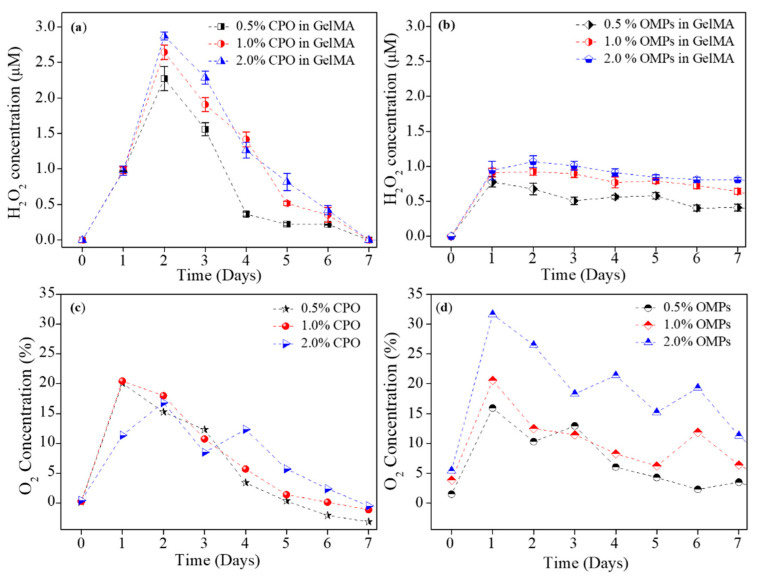
Release kinetics for H_2_O_2_ (**a**,**b**) and O_2_ (**c**,**d**) from CPO and OMPs, respectively showing the effect on hydrolysis, and hence H_2_O_2_ and O_2_ release as a function of PCL encapsulation.

**Figure 4 jfb-12-00030-f004:**
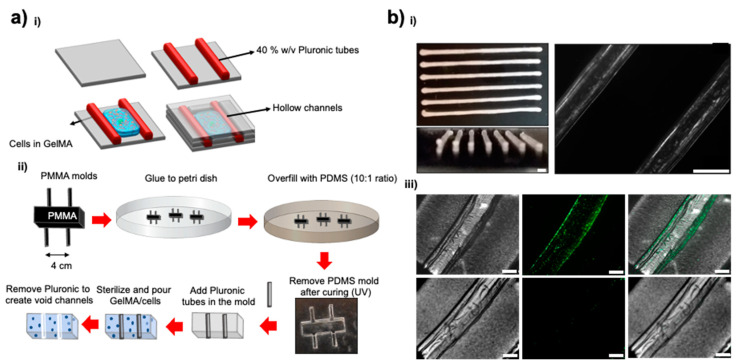
(**a**) Schematics of the fabrication of sacrificial vessels inside cellularized hydrogels; (**b**) Brightfield and fluorescent images of the fabricated hollow channel showing the inner diameter of the vessel and the before and after images of the vessel post PF-127 dissolution in water. Scale bars in (**b**) = 500 µm.

**Figure 5 jfb-12-00030-f005:**
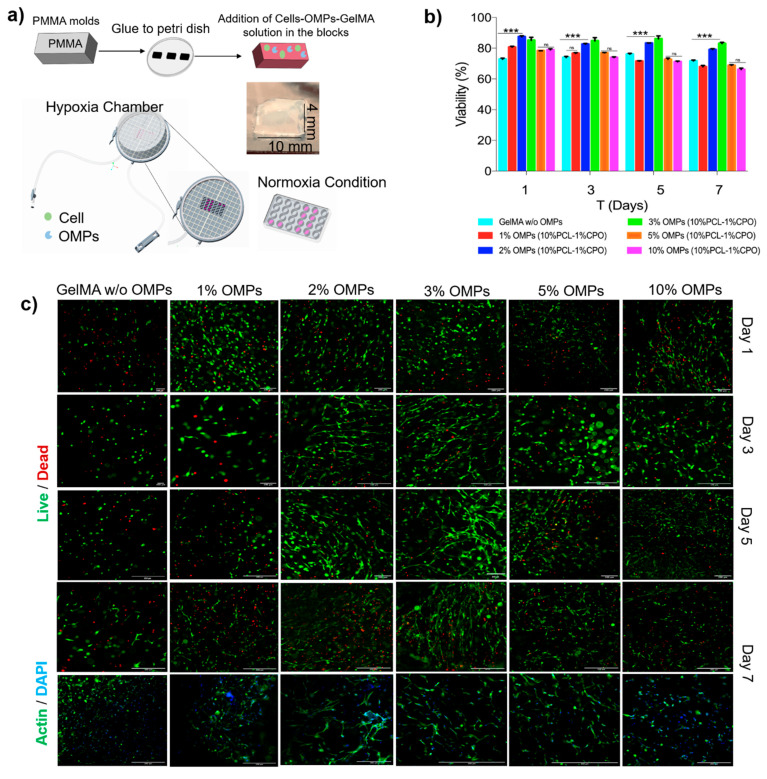
(**a**) Schematics of encapsulation of the C2C12 cells within OMPs laden hydrogels and culture under hypoxia (**b**) Cell viability data through 7 days of the culture of C2C12 cells growing in hydrogels fabricated with varying concentrations of OMPs. The lowest panel shows 7-day data on cell proliferation via actin/DAPI staining. Data are expressed as mean ± SD (n = 3). A one-way ANOVA was performed: *** *p* = 0.0007 (**c**). Representative Live/Dead and actin/DAPI images of cells/GelMA growing in hydrogels containing 1–10% OMPs. Scale bar = 200 μm.

## Data Availability

All raw data from characterization are available from the corresponding author upon request.

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
