# Peer review of "Survival and Proliferation under Severely Hypoxic Microenvironments Using Cell-Laden Oxygenating Hydrogels"

_jfb, 2021, doi:10.3390/jfb12020030_

Round 1

Reviewer 1 Report

This manuscript reports the design, formulation and characterization of oxygen releasing microparticles obtained from polycaprolactone (PCL) and calcium peroxide (CPO). The functional properties of these microparticles are assessed in cell laden hydrogels using a purposely developed microfluidic device. The results are promising and the microparticles can be proposed to enhance vascularization for the regeneration of damaged tissue.

Considering the continuous seek for new biomaterials in the field of regenerative medicine, the subject of the manuscript is of great interest. The presented results are interesting and could deserve publication.

However, the quality of the manuscript at the present stage is very low and for this reason I do not support publication in the Journal of Functional Biomaterials. A largely rewritten version of the paper could be eventually re-considered in the future.

  1. In all the sections (and specifically in the abstract, the introduction and the results and discussion) the text is confusing. The concepts are not exposed in the correct order and some of them are repeated twice with no evident reason. For these reasons, the manuscript is not effective in conveying the information as it should be.
  2. The role played by the various polymers mentioned in the text is hardly understandable. In most cases the authors refer to the figures, which only give some indications and in some cases are really not clear. As an example, the drawing at the top of Figure 1-II is not clearly connected with other parts of the scheme. Figure 2-B is also difficult to understand in the lack of any explanation in the text. Moreover, reason for the choice of the contents of the various polymers is not satisfactorily explained.
  3. The equations of Michaelis and Menten model is present in subsection 2.4. in a quite naïve way.
  4. The figures are not correctly numbered (there are two figures 3). This is a further evidence that too little attention was paid in writing the manuscript.
  5. Comments of the results presented in the figures are very poor.
  6. The conclusions section should be rewritten to clarify the advancements this work brings with respect to previous knowledge, to concisely state a “take-home” message, and to indicate directions for further research.

The text contains various errors and should be carefully revised. Below, I list just some of the first errors I found.

line 26: "alternate" should be "alternative"

line 28: "provide " should be "to provide"

line 31: "oxygen microparticles" are not defined. The authors should specify what they mean.

line 47: "This"? What does this word refer to?

In the materials and methods section the authors should report details (city, nation) of all the suppliers of chemicals and instruments.

Author Response

This manuscript reports the design, formulation and characterization of oxygen releasing microparticles obtained from polycaprolactone (PCL) and calcium peroxide (CPO). The functional properties of these microparticles are assessed in cell laden hydrogels using a purposely developed microfluidic device. The results are promising and the microparticles can be proposed to enhance vascularization for the regeneration of damaged tissue.

  1. Considering the continuous seek for new biomaterials in the field of regenerative medicine, the subject of the manuscript is of great interest. The presented results are interesting and could deserve publication.

Author Response: We appreciate this kind response from the reviewer.

  1. However, the quality of the manuscript at the present stage is very low and for this reason I do not support publication in the Journal of Functional Biomaterials. A largely rewritten version of the paper could be eventually re-considered in the future.

Author Response: All the manuscript has been rewritten and modified accordingly. We believe this version is clearer to the reader and explains the results in their proper context. We believe that the revised manuscript is much more precise and robust. We thank the editor and the anonymous reviewers for their helpful comments.   

  1. In all the sections (and specifically in the abstract, the introduction and the results and discussion) the text is confusing. The concepts are not exposed in the correct order and some of them are repeated twice with no evident reason. For these reasons, the manuscript is not effective in conveying the information as it should be.

Author Response: We sincerely accept the reviewer's input, a comprehensive review of our article and detailed suggestions. We updated the manuscript in response to the reviewers' remarks. The manuscript is now in a better shape. We are pleased that we have updated our revised manuscript in line with these comments, which has led to significant changes. Changes are listed in blue font in the revised manuscript.

  1. The role played by the various polymers mentioned in the text is hardly understandable. In most cases the authors refer to the figures, which only give some indications and in some cases are really not clear. As an example, the drawing at the top of Figure 1-II is not clearly connected with other parts of the scheme. Figure 2-B is also difficult to understand in the lack of any explanation in the text. Moreover, reason for the choice of the contents of the various polymers is not satisfactorily explained.

 Author Response: As mentioned in the other responses, we have changed the text and made respective changes in the figures to deliver the message of the paper clearer. We hope our changes address all the concerns raised.

  1. The equations of Michaelis and Menten model is present in subsection 2.4. in a quite naïve way.

 Author Response: Agreed and changed accordingly. We present the Line-weaver Burk plot that was used to actually calculate the respective release percentages.

  1. The figures are not correctly numbered (there are two figures 3). This is a further evidence that too little attention was paid in writing the manuscript.

Author response: We thank the reviewer for this advice. We have amended the mistake.

Reviewer 2 Report

This manuscript seems interesting in its conception, with two great objectives to be solved (I, II). A well-described methodology, already carried out previously, leaves some inaccuracies about the type or molecular mass of the PEG used (PEG 35kDa or PEG 400). Nor do I understand how SEM photos are taken at different percentages than the CPO content (5%) is then analyzed in release (0.5-2%), it is not explained anywhere.

However, the biggest criticism of the work is that it is disjointed. The results and discussion section is poorly presented, it is not understood, it looks like if I had used an earlier version. There is a duplication of figures 3, figure 4 is not discussed (better, it is not named) and there is a literal identical paragraph (246-249). I cannot evaluate a work that is not presented properly.

Author Response

  1. This manuscript seems interesting in its conception, with two great objectives to be solved (I, II). A well-described methodology, already carried out previously, leaves some inaccuracies about the type or molecular mass of the PEG used (PEG 35kDa or PEG 400). Nor do I understand how SEM photos are taken at different percentages than the CPO content (5%) is then analyzed in release (0.5-2%), it is not explained anywhere.

Author response: We thank the reviewer for this observation. We have made the respective changes. It should read PEG400 and CPO (1%), respectively.

  1. However, the biggest criticism of the work is that it is disjointed. The results and discussion section is poorly presented, it is not understood, it looks like if I had used an earlier version. There is a duplication of figures 3, figure 4 is not discussed (better, it is not named) and there is a literal identical paragraph (246-249). I cannot evaluate a work that is not presented properly.

Author Response: As mentioned in the other responses, we have changed the text and made respective changes in the figures to deliver the message of the paper clearer. We hope our changes address all the concerns raised.

Reviewer 3 Report

The aim of this paper was to demonstrated that 1) a new type of OMPs use to provide sufficient oxygen source to maintain healthy cellular function; (2) a model system based on 3D GelMA hydrogels containing a high cell density and oxygen generating can determine the functional effects of oxygen generating biomaterials. Even though the data is pretty interesting, the paper needs considerable clarification before it should be considered for publication.

Specific comments are below:

-The manuscript must be reorganized because it is not easy to read and to obtain the clear information. In addition, please carefully checked the figure numbers and related discussions because there are many errors in the text.  

-The authors have to describe more detailed information about the procedure of preparation materials, such as what is PCL 10% / CPO 5% / PEG 0.5%, PCL 10% / CPO 5% / Pluronic 0.5%? and what is role of GelMA in fabrication OMPs procedure shown in Figure 1(II)?

-Figure 1 Ib: MPs+GelMA sould be OMPs+GelMA

-In section 2.2, “Electrospraying (ES) polymer solution (10% PCL:1%CPO)”, this condition did not match the information shown in figure 3.

-What is the different meaning between E0 and E0 in Equation 4?

-What is the concentration of OMPs in GelMA? And how to determine it?

-L211, “… SEM (low and high magnification) images, all samples have particles with a rounded shape and porous surface”, however, except 3a and 3f, I cannot observe the obvious porous surface from figure 3. The authors should replace other clearer images.

- L223, “the nucleation and growth mechanisms of the particles are not affected when prepared under the conditions presented here”, I disagree with the statement about the particles formation. If particles were prepared by electrospraying, the particle formation was due to solvent evaporated and solidified before reaching the collector.  The mechanism is different from the nucleation and growth.

-There are many sentences in the second paragraph of the results and discussion that have been shown in previous section, such as L233-241 are same as L66-72. L247-249 repeated sentences. Please read and amend this paragraph.

- “As shown in Figure 3a, the microfluidic encapsulation block is composed of syringe 261 pumps, a collection vessel…”, but Figure 3 is SEM and particle size analysis from OMPs.

-L368, “Encapsulation of CPO in a hydrophobic polymer extends release times and lower the toxicity of CPO.” I cannot very clear what is the meaning of lower the toxicity since the figure 4 represented kinetics for H2O2 and O2 concentration.

The aim of this paper was to demonstrated that 1) a new type of OMPs use to provide sufficient oxygen source to maintain healthy cellular function; (2) a model system based on 3D GelMA hydrogels containing a high cell density and oxygen generating can determine the functional effects of oxygen generating biomaterials. Even though the data is pretty interesting, the paper needs considerable clarification before it should be considered for publication.

Specific comments are below:

-The manuscript must be reorganized because it is not easy to read and to obtain the clear information. In addition, please carefully checked the figure numbers and related discussions because there are many errors in the text.  

-The authors have to describe more detailed information about the procedure of preparation materials, such as what is PCL 10% / CPO 5% / PEG 0.5%, PCL 10% / CPO 5% / Pluronic 0.5%? and what is role of GelMA in fabrication OMPs procedure shown in Figure 1(II)?

-Figure 1 Ib: MPs+GelMA sould be OMPs+GelMA

-In section 2.2, “Electrospraying (ES) polymer solution (10% PCL:1%CPO)”, this condition did not match the information shown in figure 3.

-What is the different meaning between E0 and E0 in Equation 4?

-What is the concentration of OMPs in GelMA? And how to determine it?

-L211, “… SEM (low and high magnification) images, all samples have particles with a rounded shape and porous surface”, however, except 3a and 3f, I cannot observe the obvious porous surface from figure 3. The authors should replace other clearer images.

- L223, “the nucleation and growth mechanisms of the particles are not affected when prepared under the conditions presented here”, I disagree with the statement about the particles formation. If particles were prepared by electrospraying, the particle formation was due to solvent evaporated and solidified before reaching the collector.  The mechanism is different from the nucleation and growth.

-There are many sentences in the second paragraph of the results and discussion that have been shown in previous section, such as L233-241 are same as L66-72. L247-249 repeated sentences. Please read and amend this paragraph.

- “As shown in Figure 3a, the microfluidic encapsulation block is composed of syringe 261 pumps, a collection vessel…”, but Figure 3 is SEM and particle size analysis from OMPs.

-L368, “Encapsulation of CPO in a hydrophobic polymer extends release times and lower the toxicity of CPO.” I cannot very clear what is the meaning of lower the toxicity since the figure 4 represented kinetics for H2O2 and O2 concentration.

Author Response

The aim of this paper was to demonstrate that 1) a new type of OMPs use to provide sufficient oxygen source to maintain healthy cellular function; (2) a model system based on 3D GelMA hydrogels containing a high cell density and oxygen generating can determine the functional effects of oxygen generating biomaterials. Even though the data is pretty interesting, the paper needs considerable clarification before it should be considered for publication.

Specific comments are below:

-The manuscript must be reorganized because it is not easy to read and to obtain the clear information. In addition, please carefully checked the figure numbers and related discussions because there are many errors in the text.  

Author Response: As mentioned in the other responses, we have changed the text and made respective changes in the figures to deliver the message of the paper clearer. We hope our changes address all the concerns raised.

-The authors have to describe more detailed information about the procedure of preparation materials, such as what is PCL 10% / CPO 5% / PEG 0.5%, PCL 10% / CPO 5% / Pluronic 0.5%? and what is role of GelMA in fabrication OMPs procedure shown in Figure 1(II)?

 Author Response: We have updated the text to be more informative on how the synthesis of the particles was done. We hope it is clearer now.

-Figure 1 Ib: MPs+GelMA sould be OMPs+GelMA

  Author Response: We thank the reviewers for this comment. We have updated the text accordingly.

-In section 2.2, “Electrospraying (ES) polymer solution (10% PCL:1%CPO)”, this condition did not match the information shown in figure 3.

   Author Response: We thank the reviewers for this comment. We have updated the text accordingly.

-What is the different meaning between E0 and E0 in Equation 4?

 Author Response: Agreed and changed accordingly. We present the Line-weaver Burk plot that was used to actually calculate the respective release percentages.

-What is the concentration of OMPs in GelMA? And how to determine it?

    Author Response: We thank the reviewers for this comment. We have updated the text accordingly. The OMP concentration in GelMA was varied from 1-10% and was used as w/v for solid OMPs to liquid hydrogel.

-L211, “… SEM (low and high magnification) images, all samples have particles with a rounded shape and porous surface”, however, except 3a and 3f, I cannot observe the obvious porous surface from figure 3. The authors should replace other clearer images.

Author Response: The authors are very grateful for this referee's comment. The porosity was non evidenced for all electrosprayed OMPs groups. For this reason, the authors have changed the description in the revised version. The following word was added “partially” porous surface in the results and discussion section.

- L223, “the nucleation and growth mechanisms of the particles are not affected when prepared under the conditions presented here”, I disagree with the statement about the particles formation. If particles were prepared by electrospraying, the particle formation was due to solvent evaporated and solidified before reaching the collector.  The mechanism is different from the nucleation and growth.

 Author Response: We appreciate this comment and agree with the reviewer. We have changed the text and made respective changes in the figures as well as the text to deliver the message of the paper clearer. We hope our changes address all the concerns raised.

-There are many sentences in the second paragraph of the results and discussion that have been shown in previous section, such as L233-241 are same as L66-72. L247-249 repeated sentences. Please read and amend this paragraph.

Author response: Thanks for the comments. We have amended the sentences and also rewritten most of it anew.

- “As shown in Figure 3a, the microfluidic encapsulation block is composed of syringe 261 pumps, a collection vessel…”, but Figure 3 is SEM and particle size analysis from OMPs.

   Author Response: We thank the reviewers for this comment. We have updated the text and figure description accordingly.

-L368, “Encapsulation of CPO in a hydrophobic polymer extends release times and lower the toxicity of CPO.” I cannot very clear what is the meaning of lower the toxicity since the figure 4 represented kinetics for H2O2 and O2 concentration.

   Author Response: We thank the reviewers for this comment. The cytotoxicity comes from the bulk release of H2O2 from CPO. Encapsulation within PCL extends the release from bulk to sustained which gives enough time for its hydrolysis without affecting the cells and hence decreased cytotoxicity. We have updated the text and figure description accordingly.

Round 2

Reviewer 1 Report

I have carefully read the revised manuscript. The authors considered my indications and the text has been significantly improved. I support the publication of the manuscript in the present form.

Author Response

The authors are very gratefull for the final decision from the referee.

Reviewer 2 Report

I agree with the corrections and now the article is better understood. Just indicate that the formats of the bibliography and the draft require a deep review.

Author Response

The authors already corrected all the reference in the revised version.

Reviewer 3 Report

The authors have answered most of my questions that I pointed out in previous review report. However, the manuscript in its current form still requires amendments before being deemed publishable by Journal of Functional Biomaterials.

-L204_”The reason for this is the change in electrostatic charges and effect in weak intermolecular hydrophilic interactions that these groups affect or induce for groups that are otherwise majorly hydrophobic, such as PCL.”, I don’t know the meaning of this sentence. Maybe give more detail explanation.

- The authors still did not describe more detailed information about the procedure of preparation materials, such as what is PCL 10% / CPO 5% / PEG 0.5%, PCL 10% / CPO 5% / Pluronic 0.5%? In section 2.2, “ Electrospraying (ES) polymer solution (10% PCL:1%CPO)”, there is no mentioned about PCL 10% / CPO 5% / PEG 0.5%, PCL 10% / CPO 5% / Pluronic 0.5%

-What is role of GelMA in fabrication OMPs procedure shown in Figure 1(b)?

-Figure 1 aii: MPs+GelMA sould be OMPs+GelMA,the authors only corrected the  legend description, not the diagram. Please check the manuscript carefully.

- “All the different samples show round shaped particles with a partially porous surface”, I disagree with the use of “partially porous surface” to describe the particle surface because readers cannot observe porous from SEM images.

-  L223, the authors didn’t amend text nor did answer about the question about “the nucleation and growth mechanisms of the particles” in my previous review report.

-“L229_“An exciting approach in the field of biomaterial-based therapies is the development of oxygen-generating biomaterials. Approaches such as calcium peroxide-based oxygen generation proved capable of sustaining the cells’ metabolic needs under anoxic conditions; their unrestrained oxygen generation or release detrimentally affected the stability of the consequent oxygen tensions [1,14].”

L64_“An exciting approach in the field of biomaterial-based therapies is the development of OMPs. Platforms such as calcium peroxide (CPO)-based oxygen generation have been exploited sustaining the cells’ metabolic needs under anoxic conditions; however, the unrestrained oxygen generation and release detrimentally affected the consequent oxygen tensions' stability

The two paragraphs almost the same.

-L269 “In most cases, cell viability increased with increasing OMP concentration over a week of cell culture time under 1% O2 concentration”, I disagree with the conclusion because the cellular viability of 5% OMPs and 10% OMPs is lower than 3% on day 5 and day 7.

-L281_“achieving confluence after 21 days of culture to fill the entire volume of the tubular walls and create an interconnected vessel-like structure as a delicate equilibrium between hydrogel deterioration and cell proliferation”, there is no results related to this sentence.

Author Response

Attached file.
